# Live Cell Imaging by Förster Resonance Energy Transfer Fluorescence to Study Trafficking of PLGA Nanoparticles and the Release of a Loaded Peptide in Dendritic Cells

**DOI:** 10.3390/ph16060818

**Published:** 2023-05-31

**Authors:** Mengshan Liu, Chun Yin Jerry Lau, Irene Trillo Cabello, Johan Garssen, Linette E. M. Willemsen, Wim E. Hennink, Cornelus F. van Nostrum

**Affiliations:** 1Department of Pharmaceutics, Utrecht Institute for Pharmaceutical Sciences, Utrecht University, 3584 CG Utrecht, The Netherlands; m.liu2@uu.nl (M.L.); jerry.lau@bath.edu (C.Y.J.L.); irenichus26@gmail.com (I.T.C.); w.e.hennink@uu.nl (W.E.H.); 2Department of Pharmacology, Utrecht Institute for Pharmaceutical Sciences, Utrecht University, 3584 CG Utrecht, The Netherlands; j.garssen@uu.nl (J.G.); l.e.m.willemsen@uu.nl (L.E.M.W.); 3Department of Immunology, Nutricia Research B.V., 3584 CT Utrecht, The Netherlands

**Keywords:** cyanine-3, cyanine-5, Förster resonance energy transfer, dendritic cells, poly(lactic-*co*-glycolic acid) nanoparticles, peptide delivery

## Abstract

Our previous study demonstrated that a selected β-lactoglobulin-derived peptide (BLG-Pep) loaded in poly(lactic-*co*-glycolic acid) (PLGA) nanoparticles protected mice against cow’s milk allergy development. However, the mechanism(s) responsible for the interaction of the peptide-loaded PLGA nanoparticles with dendritic cells (DCs) and their intracellular fate was/were elusive. Förster resonance energy transfer (FRET), a distance-dependent non-radioactive energy transfer process mediated from a donor to an acceptor fluorochrome, was used to investigate these processes. The ratio of the donor (Cyanine-3)-conjugated peptide and acceptor (Cyanine-5) labeled PLGA nanocarrier was fine-tuned for optimal (87%) FRET efficiency. The colloidal stability and FRET emission of prepared NPs were maintained upon 144 h incubation in PBS buffer and 6 h incubation in biorelevant simulated gastric fluid at 37 °C. A total of 73% of Pep-Cy3 NP was internalized by DCs as quantified using flow cytometry and confirmed using confocal fluorescence microscopy. By real-time monitoring of the change in the FRET signal of the internalized peptide-loaded nanoparticles, we observed prolonged retention (for 96 h) of the nanoparticles-encapsulated peptide as compared to 24 h retention of the free peptide in the DCs. The prolonged retention and intracellular antigen release of the BLG-Pep loaded in PLGA nanoparticles in murine DCs might facilitate antigen-specific tolerance induction.

## 1. Introduction

Inducing tolerance against food allergy requires the exposure of the intestinal immune system to specific antigens from food proteins. The intestinal mucosal immune system is the largest immune system in vertebrates and the ideal target site due to its physiological propensity in distinguishing and inducing both local and systemic tolerance to orally administered harmless food proteins [1]. Dendritic cells (DCs) are regarded as the major antigen-presenting cells to contribute to tolerance induction, as they regularly sample antigens from the gastrointestinal environment for presentation to naïve T-cells and instruct adaptive immune response or tolerance thereafter [2]. Indeed, van Esch et al. [3] previously reported that oral pre-exposure to partially hydrolyzed whey protein prior to whey sensitizations increased the percentage of regulatory T-cells (Treg) in the mesenteric lymph nodes, leading to a remarkedly reduced acute allergic skin response to whole whey protein in a murine prophylactic cow’s milk allergy (CMA) model.

Orally delivered peptides are susceptible to degradation due to the acidic pH and proteolytic enzymes in the stomach, which reduce the amount of intact antigen presented to intestinal DC and thus compromise the tolerogenic outcome. Previously, Meulenbroek et al. [4] reported that oral administration of a dose of 4 mg β-lactoglobulin-derived peptides (BLG-peptides) prior to whey sensitizations reduced the acute allergic skin response to whole whey protein in mice. To improve the tolerogenic efficacy, we reported that oral pre-exposure to poly(lactic-*co*-glycolic acid) (PLGA) nanoparticles (NPs) encapsulating 160 µg BLG-peptides, but not an equivalent dose of BLG-peptides in soluble form, was effective in preventing the development of whey-induced allergy and induced systemic specific tolerance in a prophylactic murine model [5]. This result suggests that PLGA NPs play a dominating role in the improvement of the bioavailability of the encapsulated peptide cargo by its protection against proteolytic degradation in the stomach [6].

Despite the fact that antigen-loaded PLGA NPs have been investigated extensively to trigger immune responses [7,8,9,10,11], only a few studies have addressed their integrity at an acidic pH and under proteolytic conditions in the gastrointestinal tract and their internalization and subsequent intracellular trafficking in dendritic cells. Meulenbroek et al. [4] reported that the T-cell epitope containing BLG-Pep (AASDISLLDAQSAPLRVY), from a selection of β-lactoglobulin-derived peptides (BLG-peptides) of 18 amino acids (AAs), was recognized by a human T-cell line in vitro and instructed whey-specific tolerance in mice. More recently, Gouw et al. [12] identified that the BLG-Pep (AASDISLLDAQSAPLRVY) contains a sequence that can be presented by the most predominant human MHCII isotype HLR-DRB1 molecule [13]. In other words, the BLG-Pep (AASDISLLDAQSAPLRVY) with an MHCII-restricted T-cell epitope can be presented by human DCs after binding to the MHCII groove. As reviewed by Holland et al. [14], the *open-ended* conformation of the MHCII binding groove consists of a core binding 9 AAs, which allows binding to peptides of variable lengths (ranging between 12 and 20 AAs) [15]. MHCII-restricted peptides contain a central binding motif in its core, while the amino acids of outside this core peptide region are called peptide flanking regions (PFRs) [14]. It was shown that the peptide backbone of the central motifs binds to the MHCII groove by hydrogen bonding [14]. Upon the internalization of exogenous peptides by DC, MHCII molecules synthesized in the endoplasmic reticulum (ER) are transported by exocytotic vesicles containing exogenous proteins [16]. Thereafter, the acidic pH and chaperone protein HLA-DM facilitate the binding of endosomal proteolyzed MHCII-restricted peptides to the MHCII groove [17,18]. In this process, MHCII-restricted peptides undergo peptide editing mediated by HLA-DM and peptide trimming, which determines the length of the PFRs [19,20]. Hence, in the present study, we aimed to investigate the fate of internalized PLGA NPs loaded with BLG-Pep (AASDISLLDAQSAPLRVY) by murine DC 2.4 cells and the subsequent release of the loaded peptide.

To this end, we applied Förster resonance energy transfer (FRET) imaging using dual-labeled nanocarriers, which consist of donor dye-Cy3-conjugated BLG-Pep encapsulated in PLGA NP labeled with the acceptor dye Cy5 covalently linked to PLGA. Förster resonance energy transfer (FRET) is a non-radioactive energy transfer process between a pair of fluorescent probes, transferring the excess energy from the excited donor fluorochrome to the ground state acceptor by dipole–dipole coupling [21,22]. Except for the extensive overlap between the donor emission and acceptor excitation wavelengths, FRET efficiency depends predominantly on the distance between the donor and acceptor, ranging typically between 1 and 10 nm to an inverse-sixth power [21]. This strong distance-dependent characteristic enables the investigation of processes that nanoparticles undergo once taken up by cells [21]. FRET has therefore been employed as a tool to obtain insights into the degradation of internalized nanoparticles, cargo release, and particle interaction with cells [23,24,25,26]. Recent studies [25,27] reported the feasibility of obtaining insight into the intracellular fate of nanoparticles by FRET analysis by means of the co-encapsulation of both FRET donor and acceptor dyes non-covalently in PLGA NPs. Further, Zhang et al. [26] investigated the in vivo integrity of polymeric micelles by means of dual labeling of only the nanocarrier with a FRET pair dye. However, we hypothesized that the labeling of both the peptide cargo and PLGA nanocarriers in a covalent way could provide more valuable insight into the internalized PLGA nanocarriers and intracellular release of peptide payload. To this end, the peptide cargo was labeled with donor Cy3 and the PLGA nanocarrier with acceptor Cy5, and we analyzed the FRET emission upon internalization by dendritic cells. This could prove to be a biometric tool for understanding the intracellular release of loaded peptides from PLGA nanocarriers in DCs.

## 2. Results and Discussion

### 2.1. Preparation and Characterization of the Cy3-Labeled β-Lactoglobulin-Derived Peptide and Cy5-Labeled PLGA

The Pep-Cy3 (Figure 1B) was prepared by the solid phase method [28] and purified by preparative HPLC. Ultra-performance liquid chromatography (UPLC) analysis demonstrates that the purified peptide is predominantly a single entity, using UV and fluorescence detection (98.6% purity, Figure 1C). The high-resolution mass spectrometry (MS) analysis (Figure 1D) showed masses of (M^2+^) = 1172.6 and (M^3+^) = 782.1 for the purified Pep-Cy3, which is in accordance with its theoretical mass of 2344.6 g/mol. This result demonstrates the successful synthesis and purification of the Pep-Cy3. Of note, the modification of BLG-Pep by dye conjugation would inevitably influence its physicochemical properties. Importantly, the conjugation of a fluorescent dye to the N- or C-terminus of the peptide sequence will alter the resultant charge of the peptide (+1 or −1, respectively), which might influence intracellular processing. Therefore, the hydrophobic aliphatic amino acid leucine located in the middle of the BLG-peptide was substituted by a lysine-Cy3 residue, without alteration of the peptide backbone or overall charge.

### 2.2. Characteristics of Single- and Dual-Labeled PLGA Nanoparticles

The average NP diameter of fluorescent NPs ranged between 340 and 620 nm (Table 1). Positive correlations were found between fluorescence intensities and weight % of PLGA-Cy5 or Pep-Cy3, as well as the concentrations of the PLGA-Cy5 NPs and Pep-Cy3 NPs suspensions (Appendix A). The fluorescence intensity of Cy3 or Cy5 leveled off with increasing particle concentration (Appendix A) likely due to self-quenching. As reported in previous studies [25,30], this self-quenching phenomenon is dye-concentration-dependent. To avoid this self-quenching, NPs prepared with a fixed amount of 1 wt% Pep-Cy3 in the feed were selected for the preparation of FRET NPs containing 1.5–9 wt% PLGA-Cy5, corresponding to molar feed ratios of donor dye (Cy3) /acceptor dye (Cy5) (D/A) ranging from 5:1 to 0.8:1. Furthermore, to reduce the particle size, the PLGA concentration in DCM was lowered from 50 to 40 mg/mL, which decreased the average diameter of the obtained NPs from 503 to 342 nm, except for the 7 and 9 wt% PLGA-Cy5 NP (Table 1). The latter NPs had significantly larger diameters of 600–631 nm (with PDI of 0.21-0.29). Nanosight measurements showed significantly smaller mean diameters of the NP, i.e., 361 nm and 170 nm, for the particles with 7 and 9 wt% PLGA-Cy5 NP, respectively, while FRET NP 0.8:1 containing an equivalent weight percentage of PLGA-Cy5 (9%) showed a similar NP size of ~230 nm when using DLS and Nanosight measurements (Table 1). Therefore, the discrepancy in the NP size of 7 and 9 wt% PLGA-Cy5 NP might be attributed to their size distribution that more strongly influences the DLS outcome than for Nanosight [31]. Consistent with our previous study [5], the zeta potentials of the fluorescent NPs (Table 1) ranged from −0.5 to −4.7 mV, demonstrating that they all have close-to-neutral surface charges.

The encapsulation efficiency and loading capacity of the NPs increased with increasing feed ratios from 31 to 51% and 0.16 to 1.02%, respectively (Table 1). Similarly, the FRET NPs (Table 1) showed encapsulation efficiencies of the Pep-Cy3 ranging from 16 to 70%, with corresponding loading capacities ranging from 0.16 to 0.70%. The obtained peptide encapsulation and loading efficiencies are in agreement with our previous study [5].

### 2.3. Optimization of FRET Efficiency of the NPs Labeled with the Cy3 and Cy5

The spectra shown in Figure 3 were obtained by irradiating 0.5 mg/mL fluorescent NPs suspensions, namely the single-dye-labeled NPs (labeled with only Pep-Cy3 or PLGA-Cy5) and the FRET NPs (dual-labeled with both Pep-Cy3 and PLGA-Cy5), in PBS (pH 7.4) at the excitation wavelength of the donor dye Cy3 at λ_ex_ = 555 nm. As expected, 1 wt% Pep-Cy3 NP and 9 wt% PLGA-Cy5 NP showed low FRET emission at 662 nm upon excitation at 555 nm (i.e., red and black curves in Figure 3, respectively). Importantly, FRET peaks were clearly observed for Pep-Cy3 loaded in Cy5-labeled NPs prepared with different feed ratios of donor-dye-labeled peptide cargo and acceptor-dye-labeled PLGA (see FRET NP in Figure 3) upon excitation at 555 nm and emission at 662 nm.

To compare the ratiometric FRET efficiency of the FRET NPs of different feed Cy3/Cy5 molar ratios, the FRET proximity ratio (*E*_PR_) was calculated according to the Equation (1) [32]:(1)$$E_{PR}=\frac{I_{A}}{I_{A}+I_{D}},$$

Herein, $I_{A}$ is acceptor dye Cy5 emission fluorescence intensity, and$\;I_{D}$ is donor dye Cy3 emission fluorescence intensity upon excitation of the donor dye.

As seen in Table 1, the FRET NPs showed increasing *E_PR_* with decreasing D/A molar ratio up to 60% for particles prepared with a feed ratio of 1 wt% Pep-Cy3 and 9 wt% PLGA-Cy5 (i.e., 0.8:1, feed molar ratio of D/A). According to the encapsulation efficiency of Pep-Cy3, the actual D/A molar ratio of the FRET pair (Cy3/Cy5) in the FRET NP prepared with a feed ratio of 0.8:1 was 0.56:1 (Table 1), which resembles the highest FRET efficiency from a D/A ratio of about 0.5:1 as found in a previous study [33]. It is indeed likely that the highest loading capacity of Pep-Cy3 of 0.7% and highest weight percentage of PLGA-Cy5 of 9 wt% among the PLGA NPs in FRET NP 0.8:1 of similar NP size (~250 nm) concomitantly enhanced the FRET efficiency (E_FRET_). In addition, the actual FRET efficiency E_FRET_ of FRET NP 0.8:1 was 87%, from which the average separation distance of the donor and acceptor dye was calculated as 3.6 nm (see Appendix A for the corresponding calculations). In the remainder of this paper, we used the NPs with this highest FRET efficiency, namely FRET NPs containing the 0.8:1 D/A ratio (i.e., including 1 wt% Pep-Cy3 and 9 wt% PLGA-Cy5). With the control Pep-Cy3 NPs, we refer to NPs loaded with 1 wt% Pep-Cy3, while PLGA-Cy5 NPs contained 9 wt% of the Cy5-labeled PLGA.

### 2.4. Colloidal Stability and Time-Dependent Fluorescence of Labeled PLGA NP in PBS and Biorelevant Fasted-State-Simulated Gastric Fluid

The colloidal stability (size and polydispersity) of a suspension of empty PLGA NP was determined in three different media. Neither changes in the NP size nor in polydispersity (PDI) of the PLGA NPs were observed in the different media over 168 h (Appendix AA,B, blue and red and black curves, respectively), in accordance with a previous study [34]. The fluorescence stability of Pep-Cy3 NP, PLGA-Cy5 NP, and FRET NP was determined in the same media.

Appendix A shows the fluorescence spectra upon excitation at 555 nm at t = 0 h (top panels), which all have similar shapes and intensities in the different media. Upon incubation for 144 h in PBS (Figure 4C), the FRET peak at 662 nm of the FRET NP (black line) decreased to 72% of the initial value. The kinetics of this decrease is shown in Figure 4C (black squares), which was accompanied by a simultaneous increase in Cy3 fluorescence to 153% (Figure 4A, black dots) and a drop in Cy5 fluorescence intensity to 62% (Figure 4B, black stars) of their initial values. Similar results were obtained with the single-dye-labeled NP; the Cy3 fluorescence of Pep-Cy3 NP increased as well (by 115%, Figure 4A, red dots), whereas the Cy5 fluorescence of PLGA-Cy5 NP (Figure 4B, blue stars) decreased to 65% of its initial value in PBS.

In our recent study [35], the non-labeled BLG-peptide-loaded PLGA NP showed a burst release of 17% loading, followed by a slow release to 21% of the loading during the first 7 days in PBS at 37 °C. This indicates that Pep-Cy3 in the present study, in line with the previous findings, was partly released from the PLGA NPs. Recently, Yang et al. [30] reported dequenching of Dil fluorescence mediated by the release of the Dil dye loaded in Cy5-labeled PLGA NP. In line with the study of Yang et al. [30], the observed increase in Cy3 fluorescence of both Pep-Cy3 NP and FRET NP probably points to the dequenching of Cy3 fluorescence due to the release of Pep-Cy3 from the PLGA NP. The release of Pep-Cy3 also resulted in less donor dye present in proximity (less than 10 nm) of the acceptor dye (Cy5)-conjugated PLGA matrix and explains the lower FRET fluorescence intensity over 144 h as shown in Figure 4C (black squares).

To investigate the integrity and stability of the nanoparticles in biorelevant simulated gastric fluid, FRET NP and its single-labeled controls were incubated for 6 h in FaSSGF (pH 1.6, with or without pepsin) to predict their integrity during gastric passage. Figure 5C shows that upon incubation of the NPs in FaSSGF with pepsin, the fluorescence FRET intensity of the FRET NP (black squares) decreased slightly during 6 h of incubation to 86% of the initial value, accompanied by a decrease in the fluorescence intensity of Cy3 to 70% (Figure 5A) and of Cy5 to 82% of the initial value (Figure 5B). Similarly, when incubated for 6 h in FaSSGF without pepsin (Figure 5F), the FRET fluorescence signal of the FRET NP decreased to 91% of the initial value, the fluorescence intensity of Cy3 to 74% (Figure 5D), and that of Cy5 to 85% of the initial value (Figure 5E). This means that the FRET NPs have sufficient stability to survive gastrointestinal conditions after oral administration and to reach their target, namely the intestinal dendritic cells.

### 2.5. Cellular Uptake of FRET NPs by Human Immature moDC and Their Intracellular Trafficking in Murine Dendritic Cells DC 2.4

Cellular uptake of antigen-encapsulated PLGA nanocarriers by human and murine DCs is critical for the effective modulation of the innate and adaptive immune response [8] and induction of allergen-specific tolerance [36]. Thus, we investigated the cellular uptake of FRET NP by both primary human moDC and human immature moDC cells.

Initially, we investigated the uptake of nanoparticles containing only the Cy3 label (on the peptide) by human immature moDC. Figure 6A,B show that the Pep-Cy3 NP did neither affect the cell viability nor the percentage of CD11c+CD14- moDC, as compared to control cells that were exposed to the medium only. This demonstrates that the particles have good cytocompatibility at concentrations below or equal to 0.1 mg/mL NP, as also reported by Kostadinova et al. [37]. In line with the previous finding of Waeckerle-Men et al. [38] that human immature moDC are capable of internalizing PLGA particles, flow cytometry analysis showed that 73% of the CD11c+CD14- moDC were positive for the Cy3 fluorescence signal (Figure 6C).

In parallel, confocal fluorescence microscopy (CFM) images of these cells incubated with only the medium (Figure 7A) and Pep-Cy3 NP (Figure 7B) confirmed the internalization of Pep-Cy3 NP after 2.5 h of incubation and showed even higher brightness after 18.5 h of incubation (Figure 7C).

As compared to the primary suspension culture of human moDC, the less costly and more accessible immortalized murine DC 2.4 cell line was selected for further investigations of intracellular trafficking of PLGA nanoparticles and encapsulated peptide. Murine DC 2.4 cells were incubated with PLGA-Cy5 NP, Pep-Cy3 NP, and FRET NP and imaged by CFM. It is noted that non-internalized particles of >200 nm could not be separated from the moDC by washing and centrifugation at 300× *g*. Likely, these bigger particles co-sediment with the cells during centrifugation. Therefore, the nanoparticles were filtered before exposure to the cells, using a 0.2 µm RC membrane filter. As shown in Appendix A, some fluorescence was lost after filtration of the 0.5 mg/mL Pep-Cy3 NP and FRET NP; 0.9 μg/mL of encapsulated Pep-Cy3, corresponding with particle concentrations of 0.30 and 0.14 mg/mL, respectively, remained in the samples after filtration. Therefore, for proper comparison with free Pep-Cy3 controls, we applied 1 μg/mL of the free peptide in the following experiments. Even more of the PLGA-Cy5 NP suspension was lost after filtration (Appendix A) due to their larger mean NP size (631 nm as compared to 236 nm for the FRET NP), resulting in 0.03 mg/mL PLGA-Cy5 NP, which is much lower than 0.14 mg/mL for the FRET NP. This might explain the discrepancy of Cy5 fluorescence intensity observed for cells incubated with the PLGA-Cy5 NP and FRET NP at 2 h after incubation, as shown in the top panels of Figure 8B and the zoomed-out images in Appendix AA.

First, images were taken after 2.5 h of exposure of the murine DCs with the NPs, washing, and 2 h of further incubation (top panels of Figure 8B). These images show partial accumulation and heterogeneous distribution of Cy3 fluorescence in DC 2.4 cells incubated with the free Pep-Cy3, while cells incubated with FRET NP showed a more homogeneous intracellular distribution with few separate spots of orange Cy3 fluorescence from internalized FRET NP (see also Cy3 signal only, in the left panel of Figure 8C). These spots might be attributed to the accumulation of free and encapsulated Pep-Cy3 in endosomes of DC. The uptake of soluble antigens by DC is mainly mediated by macropinocytosis [39], while the internalization of PLGA NP with an NP size of ~290 nm by the same cells is mediated by phagocytosis, as previously shown by Diwan et al. [40], which might explain the observed difference in the intracellular distribution of free Pep-Cy3 and encapsulated Pep-Cy3 in FRET NP.

It is noted that Cy3 fluorescence is diminished in FRET NP (see also Cy3 signal only, in the left panel of Figure 8C) likely due to the self-quenching and quenching effect by non-radioactive energy transfer from the emission of Cy3 to the acceptor Cy5. Indeed, Figure 9A shows Cy3 fluorescence colocalized with the FRET signal, as indicated by the arrows, suggesting the integrity of internalized FRET NP at 2 h post-incubation.

Upon continued incubation, PLGA NP might be partially degraded in the lysosomal compartments of dendritic cells 24 h post incubation [25] and/or endo-lysosomal escape [25,41,42]. At the 24 h time point (Figure 8C, right panel), the Cy3 fluorescence of FRET NP retained the spot-like pattern, albeit with lower intensity, while the Cy3 fluorescence was predominantly cleared from cells incubated with free Pep-Cy3. The observed short retention time of internalized free Pep-Cy3 is consistent with the previously reported retention time of less than 24 h for ovalbumin (OVA) taken up by dendritic cells [43,44]. To maintain intracellular homeostasis, internalized exogenous proteins and/or nanoparticles are eliminated by energy-dependent lysosome-mediated exocytosis [45,46], which was found to be faster for low-molecular-weight solutes and slower for PLGA nanoparticles [46]. In fact, Figure 8B demonstrates that the internalized FRET NP remained in the DC for at least 144 h, which demonstrates a significantly longer retention time within the cells as compared to the free Pep-Cy3 at an equivalent dose. To further explain, after internalization of the free Pep-Cy3, it is translocated from macropinosomes to early endosomes, which subsequently matured into late endolysosomes for MHCII presentation on the surface of DCs [47]. On the other hand, after phagocytosis by DC, the internalized FRET NP localized in endosomes [48], which possibly underwent a similar intracellular process as an internalized soluble peptide for MHCII presentation [49].

To focus on DC 2.4 cells incubated with FRET NP only, Figure 9 and the same but zoomed-out images in Supplementary Figure 5 show from the left to the right the separate confocal microscopy images of Cy3, Cy5, and FRET fluorescence channels and their merged confocal fluorescence microscopy images at different incubation times. Colocalization of Cy3, Cy5, and the FRET signal was observed and exemplified by the numbered arrows pointing towards the same spot in every image at each time point (2–144 h).

Live images in Figure 9B showed a considerable loss of intracellular FRET signal (purple spots) after 24 h, which may be caused by the release of Pep-Cy3 from the FRET NP. At the 96 h time point (Figure 9C), the colocalization of Cy3, Cy5, and the FRET signal was still visible, while at the same time point, full clearance of Cy3 fluorescence was observed for the free Pep-Cy3-incubated DC 2.4 cells (Appendix AC). At 144 h, although Cy5 and Cy3 fluorescence remained detectable, the FRET signal was cleared (Figure 9D and the same but zoomed-out images in Appendix AD), suggesting the quantitative intracellular release of Pep-Cy3 and/or complete degradation of the internalized FRET NP. Our findings demonstrate that PLGA nanoparticles prolong the intracellular presence of encapsulated Pep-Cy3 in dendritic cells, which might be attributed to the endosomal escape of the PLGA nanoparticles into the cytosol [48]. Given the short life span of dendritic cells of less than 9 days [50], the prolonged intracellular presence of the peptide upon incubation with peptide-loaded PLGA nanoparticles may result in a prolonged period of peptide presentation via MHCII by the DC to the T-cell receptor (TCR) of antigen-specific T-cells. This, in turn, may contribute to the efficiency of inducing peptide-specific tolerogenic T-cell responses to induce oral tolerance using a relatively low dose of the peptide.

## 3. Materials and Methods

### 3.1. Materials

Preloaded Fmoc-Tyr (tbu)-Wang resin, 9-fluorenylmethyloxycarbonyl (Fmoc)-protected amino acids, and trifluoroacetic acid (TFA) were purchased from Novabiochem GmbH (Hohenbrunn, Germany). Peptide-grade dimethylformide (DMF), dichloromethane (DCM), piperidine, *N*, *N*′-diisopropylcarbodiimide (DIC), and HPLC-grade acetonitrile were purchased from Biosolve BV (Valkenswaard, The Netherlands). Ethyl cyanohydroxyiminoacetate (Oxyma pure) was purchased from Manchester Organics Ltd. (Cheshire, UK). Triisopropylsilane (TIPS), BioUltra-grade ammonium bicarbonate, polyvinyl alcohol (PVA, 87–90% hydrolyzed, Mw 30,000–70,000 Da), sodium chloride, Dulbecco’s Phosphate Buffered Saline (DPBS), RPMI 1640 medium, Fetal Bovine Serum (FBS), *N*, *N*′-dicyclohexylcarbodiimide (DCC), *N*-hydroxysuccinimide (NHS), triethylamine (TEA), and lithium chloride (LiCl) were purchased from Sigma-Aldrich Chemie BV (Zwijndrecht, The Netherlands). Cyanine3-labeled Fmoc-Lysine (Fmoc-Lys (Cy3)-OH) was purchased from APPTec (Louisville, CT, USA). Cy5 amine was purchased from Luminprobe (Hannover, Germany). Uncapped poly (dl-lactide-*co*-glycolide) (PLGA-COOH) (PURASORB PDLG 5004A, lactide/glycolide 50/50, intrinsic viscosity 0.32–0.48 dL/g) was purchased from Corbion (Gorinchem, the Netherlands). Penicillin and streptomycin were purchased from Gibco (New York, NY, USA), and phosphate-buffered saline (10×PBS, containing 1.37 M NaCl, 0.027 M KCl, and 0.119 M phosphates) was obtained from Fisher BioReagents (Pittsburgh, PA, USA) and diluted 10 times with Milli-Q water before use. Bovine serum albumin (BSA) was obtained from Sigma-Aldrich Chemie BV (Zwijndrecht, The Netherlands). Fasted-state-simulated gastric fluid (FaSSGF) (pH 1.6, without pepsin) was prepared by solubilizing 80 µM NaTc (sodium taurocholate, Santa Cruz biotechnology, Dallas, TX, USA), 20 µM lecithin (phosphatidylcholine from egg, Liphoid GmbH, Ludwigshafen, Germany), and 34.2 µM NaCl (Sigma-Aldrich) in Milli-Q water and adjusted to pH 1.6 with 1N HCl [51]. For the preparation of FaSSGF with pepsin, 0.1 mg/mL pepsin from porcine gastric mucosa (Sigma-Aldrich) was supplemented to the FaSSGF (pH 1.6).

### 3.2. Solid-Phase Peptide Synthesis and Characterization

Cy3-labeled peptide (Pep-Cy3; structure shown in Figure 1B) was obtained using a microwave-assisted solid-phase peptide synthesis method [52] using an H12 liberty blue peptide synthesizer (CEM Corporation, Matthews, SC, USA). DMF was used as the coupling and washing solvent. For each coupling step, Fmoc-amino acids were activated by 5eq of Oxyma pure and DIC to react with the free N-terminal amino acids on the resin for 1 min at 90 °C. After each coupling step, the Fmoc groups were removed by treatment with 20% piperidine for 1 min at 90 °C. Cy3 modification of the β-lactoglobulin-derived peptide (Figure 1A) was performed by introducing Fmoc-Lys (Cy3)-OH residue to the peptide sequence. TFA/H_2_O/TIPS (95/2.5/2.5 *v*/*v*/*v*) was used to simultaneously cleave the peptide from the resin and remove the side-chain protecting groups.

Pep-Cy3 was purified by a Prep-HPLC using a Reprosil-Pur C18 column (10 μm, 250 × 22 mm) and eluted with a water-acetonitrile gradient from 5 to 80% acetonitrile (10 mM ammonium bicarbonate) in 25 min at a flow rate of 15 mL/min with UV detection at 220 nm.

The purity of the obtained Pep-Cy3 was analyzed with ultra-performance liquid chromatography (UPLC) by using a BEH C18 1.7 µm column (Waters^®^, Milford, CT, USA), UV detection at 210 nm, and fluorescence detection at λ_ex/em_ = 555/570 nm. A gradient elution method was used with a mobile phase A (0.1% TFA, 5% acetonitrile, and 95% H_2_O) and a mobile phase B (acetonitrile). The eluent changed linearly from 20 to 40% mobile phase B in 6 min with a flow rate of 0.25 mL/min at 25 °C. Mass spectrometry (MS) analysis of the synthesized Pep-Cy3 was performed using a microTOF-Q instrument in positive mode.

### 3.3. Conjugation of Cy5 to PLGA-COOH

The synthesis of PLGA-Cy5 (Figure 2A) was adapted from a method described previously [29]. PLGA-COOH (M_n_ = 13,700 g/mol according to GPC analysis using PEG standards, 211.4 mg) was dissolved in anhydrous DCM (100 mg/mL) at room temperature. Next, DCC (31.0 mg, 150 mmol) in 1 mL anhydrous DCM was added to the PLGA-COOH solution and stirred for 10 min. Subsequently, NHS (17.5 mg, 152 mmol) dissolved in 1 mL anhydrous DCM was added, and the resulting reaction mixture was stirred for 60 min at room temperature. Next, the formed white precipitate *N, N*′-dicyclohexylurea (DCU) was removed by filtration with a 0.2 μm regenerated cellulose (RC) membrane syringe filter (Phenomenex, Torrance, CA, USA). The filtrated PLGA-NHS solution was subsequently added to a mixture of triethylamine (17.3 mg, 0.084 mmol) and Cy5 amine (5.0 mg, 0.0076 mmol) in 0.5 mL anhydrous DCM and stirred for 8 h at room temperature. Next, the synthesized PLGA-Cy5 was precipitated into 80 mL diethyl ether/methanol (1:1 *v*/*v*) and centrifuged at 2700× *g* for 10 min at 4 °C. The supernatant was decanted, and the precipitated product was dried under a nitrogen atmosphere for 10 min to remove residual solvents. Next, the product was dissolved in 4 mL DMSO and transferred into a 3.5 kDa Spectra/Por RC membrane tube (Thermo Fisher Scientific, Washington, DC, USA) for dialysis against 100 mL acetonitrile. The dialysis medium was refreshed daily for 7 days. The obtained PLGA-Cy5 was precipitated by dropping the polymer solution into 100 mL milli-Q water. The formed precipitate was collected and subsequently lyophilized overnight to yield PLGA-Cy5 in the form of small blue pellets. PLGA-Cy5 was analyzed with GPC (Waters^®^ alliance e2695 system, Milford, CT, USA), which was equipped with a UV/Vis detector (Waters^®^ 2489) and refractive index detector (Waters^®^ 2414) using DMF/10 mM LiCl as the solvent, and PEG of a defined molecular weight was used as calibration standards.

### 3.4. Preparation of Nanoparticles

A double emulsion solvent evaporation method was applied to prepare empty PLGA-Cy5 NP, non-labeled PLGA NP loaded with peptide-Cy3 (Pep-Cy3 NP) and PLGA-Cy5 NP loaded with peptide-Cy3 (i.e., FRET NP) [5]. Briefly, 0.125 mL PBS (for PLGA-Cy5 NP) or PBS containing 0.04, 0.2, 0.4, 0.6, or 0.8 mg Pep-Cy3 (corresponding to 0.1, 0.5, 1, 1.5, or 2 wt% Pep-Cy3 NP or FRET NP) as the internal aqueous phase was added to 1.25 mL of anhydrous DCM in which 50 mg of a blend of PLGA-Cy5 (0, 1.5, 2, 4, 7, or 9 wt%) with PLGA-COOH was dissolved prior to sonication using a Sonifier S-450A (3 mm, Branson Ultrasonics B.V., Soest, The Netherlands) at 20% amplitude for 1 min on an ice bath to yield a water-in-oil emulsion. Next, the emulsion was added to 12.5 mL of the external aqueous phase, containing 3% *w*/*v* PVA and 0.9% *w*/*v* NaCl, and the obtained mixture was sonicated using a Sonifier S-450A (13 mm, Branson Ultrasonics B.V.) at 20% amplitude for 1 min on an ice bath to yield a water-in-oil-in-water emulsion. Finally, the formed emulsion was stirred for 4 h at room temperature to evaporate DCM and to obtain hardened PLGA NPs. The nanoparticle suspension was centrifuged at 20,000× *g* for 30 min at 4 °C, and the pelleted nanoparticles were washed with 10 mL PBS twice. The nanoparticles were resuspended in 2 mL of Milli-Q water and lyophilized using a freeze dryer (Lyovapor L-200, BUCHI Corporation, New Castle, DE, USA).

### 3.5. Characterization of the Nanoparticles

After the washing step, the nanoparticles pellet was resuspended in 2 mL of Milli-Q water (in Section 3.4), from which a 6 μL nanoparticle suspension was added to 1994 μL of Milli-Q water for the characterization of nanoparticle size and polydispersity index (PDI) using Zetasizer Nano S (Malvern Instruments, Malvern, UK).

In addition, 100 μL of 100 mM HEPES buffer (pH 7.4) was added into a 900 μL nanoparticles suspension in Milli-Q water for measurement of the zeta potential of nanoparticles, using Zetasizer Nano-Z (Malvern Instruments, Malvern, UK).

To avoid possible inaccuracy caused by the overlapping emission of the laser of the Zetasizer Nano S with Cy5 absorption [53], the size of Cy5-labeled NP was also measured using Nanosight LM14 (Malvern Instruments, Malvern, UK).

### 3.6. Peptide Encapsulation Efficiency

The encapsulation efficiency of Pep-Cy3 was determined using an indirect method [54]. The supernatants collected from the washing steps were filtered using a 0.2 μm RC syringe filter (Phenomenex) prior to Acquity ultra-performance liquid chromatography (UPLC) analysis. The Pep-Cy3 concentration of the collected filtered aqueous phases was determined using a BEH C18 1.7 µm column (Waters) and UV detection at 210 nm and fluorescence detection at λ_ex/em_ = 555/570 nm. A gradient elution method was used with mobile phase A (0.1% TFA, 5% acetonitrile, and 95% H_2_O) and mobile phase B (0.1% TFA and acetonitrile). The eluent changed linearly from 20 to 40% mobile phase B in 6 min with a flow rate of 0.25 mL/min at 25 °C.

Peptide standards (1–100 µg/mL Pep-Cy3, 7.5 µL injection volume) dissolved in mobile phase A were used for calibration. The encapsulation efficiency (EE) of Pep-Cy3 in PLGA NP is defined as the amount of encapsulated peptide (which equals the weight of peptide in the feed minus the weight of quantified non-encapsulated peptide) divided by the weight of peptide used for the preparation of the loaded NPs (Equation (2)) [54]. The loading capacity of Pep-Cy3 in PLGA NP is reported as the weight of the encapsulated peptide divided by the weight of the peptide-loaded NPs (Equation (3)) [55].
(2)$$\mathrm{Encapsulation}\;\mathrm{Efficiency}\;\left(\mathrm{EE}\right)\;\left(\%\right)=\frac{{\mathrm{Weight}}_{\mathrm{feed}\;\mathrm{peptide}}-{\;\mathrm{Weight}}_{\mathrm{non}-\mathrm{encapsulated}\;\mathrm{peptide}}\;}{{\mathrm{Weight}}_{\mathrm{feed}\;\mathrm{peptide}}}\;\times \;100\%,$$
(3)$$\mathrm{Loading}\;\mathrm{Capacity}\;\left(\mathrm{LC}\right)\;\left(\%\right)=\frac{{\mathrm{Weight}}_{\mathrm{encapsulated}\;\mathrm{peptide}}}{\mathrm{Weight}\;\mathrm{of}\;\mathrm{nanoparticles}\;}\times 100\%,$$

### 3.7. Fluorescence Characteristics of Labeled Nanoparticles

Steady-state fluorescence was measured in semi-micro cell quartz cuvettes (10 × 4 × 45 mm, Hellma™ Suprasil™, Thermo Fisher Scientific) in a PCT-818 Automatic 4-position Peltier cell changer equipped Jasco Spectrofluorometer FP-8300 (Easton, PA, USA), from 1 mL 0.5 mg/mL PLGA-Cy5 NP (containing 1.5–9 wt% of Cy5-labeled PLGA), Pep-Cy3 NP (0.1–2 wt% feed Pep-Cy3), or FRET NP (1 wt% feed Pep-Cy3, 1.5–9 wt% PLGA-Cy5) in PBS (pH 7.4). Briefly, the Cy3 fluorophore was excited at λ_ex_ = 555 nm using a wavelength width of 5 nm, and the Cy3 fluorescence was recorded at λ_em_ = 570 nm. The Cy5 fluorophore was excited at λ_ex_ = 646 nm, and fluorescence emission was recorded at λ_em_ = 662 nm, using a wavelength width of 5 nm. FRET fluorescence spectra at λ_em_ = 570–900 nm were recorded for 0.5 mg/mL FRET NP (1 wt% feed Pep-Cy3 and 1.5–9 wt% PLGA-Cy5) and PLGA-Cy5 NP as the control (containing 1.5–9 wt% PLGA-Cy5) by the excitation of Cy3 at λ_ex_ = 555 nm using a wavelength width of 5 nm.

### 3.8. Fluorescence Intensity of Labeled PLGA NP in PBS or Fasted-State-Simulated Gastric Fluid

The fluorescence intensity of PLGA-Cy5 NP (9 wt% PLGA-Cy5) and FRET NP (0.8:1, i.e., 1 wt% Pep-Cy3, 9 wt% PLGA-Cy5) was monitored upon incubation at 37 °C for 144 h in PBS or fasted-state-simulated gastric fluid (FaSSGF) [51]. Briefly, lyophilized fluorescently labeled PLGA NPs were suspended at 0.2 mg/mL in PBS (pH 7.4), FaSSGF (pH 1.6, with pepsin), or FaSSGF (pH 1.6, without pepsin). Next, 1 mL of NP samples was pipetted into semi-micro cell quartz cuvettes (10 × 4 × 45 mm, Hellma™ Suprasil™) and closed with a lid for incubation at 37 °C. At different time points (after 0.5, 2, 4, 24, 48, 120, and 144 h), the fluorescence emission of Cy5 and FRET was recorded using Jasco Spectrofluorometer FP-8300.

### 3.9. Culture of Human-Monocytes-Derived Dendritic Cells (moDC)

Freshly donated and condensed blood (buffy coat) (time between donation and experiments was 24 h at most) was obtained from the Dutch blood bank (Amsterdam, The Netherlands). A total of 25 mL of fresh human blood was diluted with the same volume of DPBS-2% FBS at room temperature and slowly transferred into Leucosep tubes (VWR, Radnor, PA, USA), followed by centrifugation for 13 min at 1000× *g* at room temperature with slow acceleration and deceleration. Subsequently, the interface containing the peripheral blood mononuclear cells (PBMCs) was centrifuged for 5 min at 1800 rpm at room temperature, and the supernatant was removed. A total of 50 mL of DPBS-2% FBS was added to resuspend the pellets, and centrifugation was repeated 4 times until the supernatant was clear. Next, 5 mL of the lysis buffer (8.3 g/L NH_4_Cl, 1 g/L KHCO_3,_ and 37 mg/L EDTA), filtered using a sterile 0.22 µm syringe filter (Cellulose Acetate, Whatman™, GE Healthcare Life Sciences, Thermo Fisher Scientific), was added to the pellets and incubated for 4 min on ice to lyse the erythrocytes. Afterward, the obtained PBMC pellet was resuspended in RPMI 1640 medium-10% FBS-1% penicillin/streptomycin and counted using a cell counter (Z1 Coulter Particle Counter, Beckman Coulter™, Indianapolis, IN, USA) prior to the isolation of monocytes by negative selection with human Monocyte Isolation Kit II and MACS column and MACS separator. Per donor, 1 × 10^6^ monocytes per 1 mL were cultured in the presence of 100 ng/mL human recombinant IL-4 and 60 ng/mL recombinant human GM-CSF to induce differentiation into immature-monocytes-derived dendritic cells (moDC) in 6-well suspension culture plates (Greiner Bio-one, Solingen, Germany) and incubated at 37 °C and 5% CO_2_. On days 3 and 6, half of the medium was withdrawn and replaced with fresh medium containing 100 ng/mL IL-4 and 60 ng/mL GM-CSF.

### 3.10. Culture of Murine Bone-Marrow-Derived Dendritic Cell Line DC 2.4

Immortalized murine bone-marrow-derived dendritic cell line DC 2.4 (Merck, Kenilworth, NJ, USA) was cultured in a T75 flask with RPMI 1640 medium-10% FBS supplemented with 10 µM β-mercapoethanol, 1 × L-glutamine solution, 1 × trypsin-EDTA solution, MEM non-essential amino acids, and 1 × HEPES solution according to the supplier’s instructions.

### 3.11. Internalization of Pep-Cy3/NP by Human moDC Determined by Fluorescence Microscopy and Flow Cytometry

A suspension of Pep-Cy3 NP (with 2 wt% feed Pep-Cy3) dispersed in RPMI 1640 medium-10% FBS-1% penicillin/streptomycin at a concentration of 1 mg/mL was filtered using a 0.2 µm RC membrane filter (Phenomenex). The concentration of the filtrated fluorescent NP suspension was determined after 5 times of dilutions in the RPMI 1640 medium-10% FBS-1% penicillin/streptomycin by measuring the Cy3 fluorescence using Jasco Spectrofluorometer FP-8300. After 7 days of differentiation as indicated in Section 3.9, immature human moDC of 3 healthy donors were seeded at a density of 2.5 × 10^5^ cells per well in a 96-well cell culture plate. On day 7, the immature moDC were incubated with medium or 0.060 mg/mL Pep-Cy3 NP (2 times diluted as indicated in Appendix A) in 250 µL RPMI 1640 medium-10% FBS-1% penicillin/streptomycin and incubated at 37 °C and 5% CO_2_ for 2.5 and 18.5 h, respectively. After incubation, the cells were transferred from the wells into a 15 mL falcon tube and washed twice with 7 mL RPMI 1640 medium-10% FBS-1% penicillin/streptomycin by using centrifugation at 300× *g* for 5 min at room temperature. Approximate 60 μL of RPMI 1640 medium-10% FBS-1% penicillin/streptomycin was added to the cells prior to Yokogawa imaging as described in Section 3.12.

In parallel, cellular uptake of Pep-Cy3 NP by the immature moDC was quantified using flow cytometry. The cells were seeded at a density of 2 × 10^5^ cells per well in a 48-well cell culture plate and subsequently incubated at 37 °C and 5% CO_2_ for 2.5 h with medium or 0.060 mg/mL Pep-Cy3 NP (2 times diluted as indicated in Appendix A) in 200 µL of RPMI 1640 medium-10% FBS-1% penicillin/streptomycin. Next, the cells were transferred into a 15 mL falcon tube and washed twice with 7 mL RPMI 1640 medium-10% FBS-1% penicillin/streptomycin by using centrifugation at 300× *g* for 5 min at room temperature. Prior to flow cytometry analysis, the viability of the cells was determined by staining them with the fixable viability dye eFluor™780 (at 2000 times dilution in DPBS) (Thermo Fisher Scientific) and incubated on ice for 30 min. To avoid non-specific binding between the Fc domain of the IgG antibodies and the Fc receptors expressed on human moDC, the cells were incubated with 25 µL 1% human Fc-blocking antibody (BD Biosciences) in DPBS-1 % bovine serum albumin (BSA)-2% FBS buffer per well for 10 min at 4 °C [56]. For characterization of human CD11c+CD14- moDC, the cells were incubated with CD11c-PerCP/eFluor 710 (at 640 times dilution) (Thermo Fisher Scientific) and CD14-eFluor450 (at 100 times dilution) (BD Biosciences) in DPBS-1 % BSA buffer for 45 min at 4 °C in the dark. Cells were analyzed with a BD FACSCantoII flow cytometer (Becton Dickinson, Franklin Lakes, NJ, USA). Data analysis was conducted with FlowLogic software (Inivai Technologies, Mentone, Australia).

### 3.12. Cellular Uptake and Intracellular Trafficking of Nanoparticles by Murine DC 2.4

Murine immortalized dendritic cells DC 2.4 were seeded at a density of 5 × 10^4^ per well and cultured in a 96-well light-proof plate for 2 days at 37 °C in 250 μL RPMI 1640 to reach a density of approximately 2 × 10^5^ cells per well. For reasons explained in Section 3.11, the different NP suspensions of PLGA-Cy5 NP (9 wt% labeled polymer), Pep-Cy3 NP (1 wt% feed Pep-Cy3), or FRET NP 0.8:1 (1 wt% Pep-Cy3:9 wt% PLGA-Cy5) dispersed in RPMI 1640 medium at a concentration of 0.5 mg/mL were filtered using a 0.2 µm RC membrane filter (Phenomenex) to remove NP with a size above 200 nm. The concentration of particles in the filtrated fluorescent NPs suspensions was determined by measuring the Cy3 and Cy5 fluorescence using Jasco Spectrofluorometer FP-8300.

DC 2.4 cells were rinsed twice with DPBS prior to the addition of 200 μL of the filtered fluorescent NP suspension or a solution of the Pep-Cy3 (concentration 1 μg/mL, equivalent concentration as in the FRET NP suspension) in RPMI 1640 medium to the wells. After 2.5 h incubation at 37 ºC, the nuclei of the DC 2.4 cells were stained with the dye Hoechst 33342 (5 μg/mL) for 15 min at 37 ºC. Subsequently, the stained cells were rinsed 3 times using 200 μL PBS to remove the extracellular dye and the non-internalized nanoparticles. Finally, live imaging of the cellular uptake and intracellular trafficking of fluorescent nanoparticles by the nuclei-stained cells over 144 h at 37 ºC in RPMI 1640 medium was conducted using a confocal laser scanning microscope system (Cell Voyager CV7000S, Yokogawa Electric, Tokyo, Japan) at excitation at 405 ± 5 nm (Hoechst), 561 ± 2 nm (Cy3 or FRET), or 640 ± 4.5 nm (Cy5). Live confocal images were acquired using a water immersion lens 60× objective and emission band-pass filters: BP525/50 (Bright field, confocal path), BP445/45 (Hoechst), BP600/37 (Cy3), and BP676/29 (Cy5 or FRET). Compensation for non-specific fluorescence of the different images was conducted for cells incubated with medium, 1 wt% Pep-Cy3 NP, 9 wt% PLGA-Cy5 NP, and FRET NP (0.8:1, 1 wt% Pep-Cy3:9 wt% PLGA-Cy5) using Columbus Image Data Storage and Analysis System (on http://columbus.science.-uu.nl) (accessed on 24 May 2023).

## 4. Conclusions

To investigate whether the β-lactoglobulin-derived peptide-loaded NPs could be internalized and processed by dendritic cells resulting in the intracellular release of the peptide, we exploited nanoparticles with high FRET emission that were obtained by dual labeling of the PLGA nanocarrier (Cy5) and the encapsulated peptide cargo (Cy3). Our study demonstrates sufficient stability and integrity of the peptide-loaded PLGA nanoparticles (FRET NP) in the simulated GI tract conditions, and very efficient cellular uptake of peptide-loaded NP by human moDC as confirmed by fluorescence microscopy imaging and flow cytometry. As compared to a 24 h retention time of the non-encapsulated peptide, prolonged retention of the encapsulated peptide cargo for 96 h by DC 2.4 was demonstrated. Collectively, we show here that NP with a dual FRET labeling strategy of both the peptide cargo and the nanocarriers provides a novel biometric tool to obtain valuable insights on the trafficking and processing of peptide-loaded NPs in DC in vitro.

## Figures and Tables

**Figure 1 pharmaceuticals-16-00818-f001:**
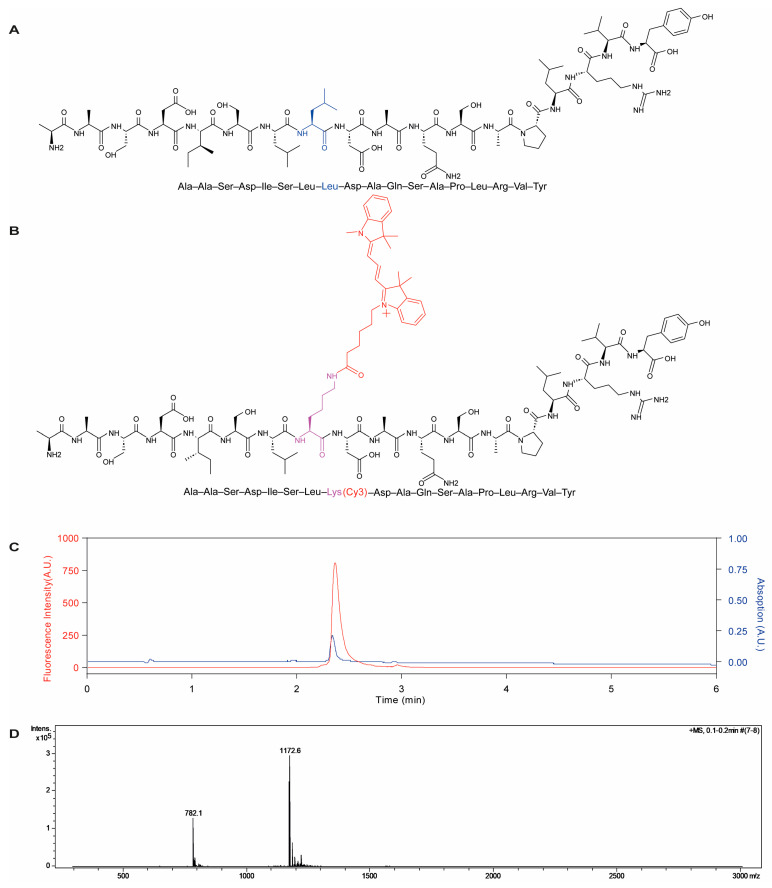
Structures of parent β-lactoglobulin-derived peptide (**A**) and Pep-Cy3 (**B**), UPLC chromatograms of Pep-Cy3 (**C**) used for determination of purity using UV detection at 210 nm (blue curve) and fluorescence detection at λ_ex/em_ = 555/570 nm (red curve) and (**D**) molecular mass of the synthesized Pep-Cy3 by mass spectrometry (Theoretical mass = 2343.3 g/mol; mass found (M2+) = 1172.6 and (M3+) = 782.1; and calculated mass = 2343.2 g/mol or 2343.3 g/mol). Cy5 was conjugated to PLGA-COOH by using carbodiimide/NHS chemistry as described previously [29] (Figure 2A). Gel permeation chromatography (GPC) of the synthesized PLGA-Cy5 showed overlapping peaks at the same retention time when using the refractive index (RI) and UV detection at 650 nm (Figure 2B,C, respectively), while no free dye was detected (Figure 2C), which demonstrates the successful conjugation of the acceptor dye Cy5 to PLGA-COOH.

**Figure 2 pharmaceuticals-16-00818-f002:**
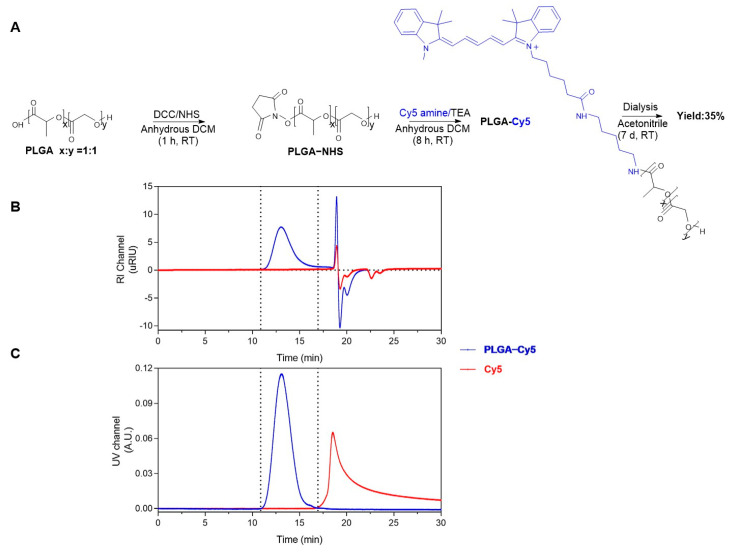
Synthesis and characterization of PLGA-Cy5. (**A**) Synthesis route of PLGA-Cy5 by conjugation of Cy5 amine to PLGA-COOH using carbodiimide/NHS chemistry. GPC analysis of PLGA-Cy5 conjugate (blue) and free Cy5 (red) by using refractive index (RI) detection (**B**) and UV detection at 650 nm (**C**).

**Figure 3 pharmaceuticals-16-00818-f003:**
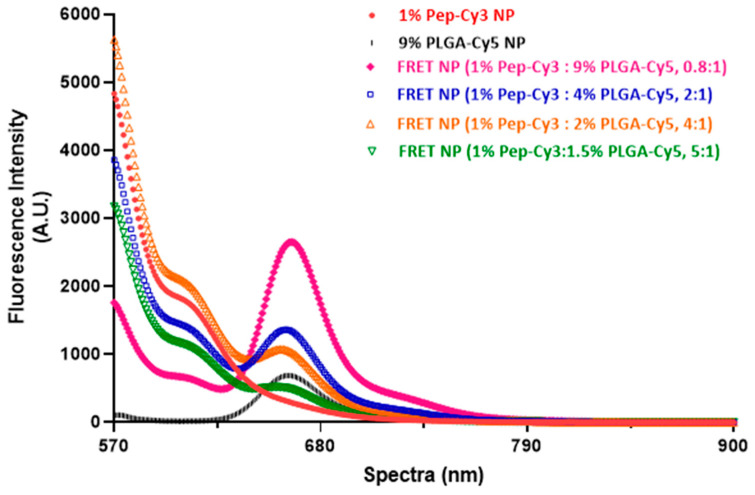
Fluorescence spectra in PBS (pH 7.4) at λ_ex_ = 555 nm: 0.5 mg/mL 9% PLGA-Cy5 NP (containing 9 wt% labeled polymer), 1% Pep-Cy3 NP (containing 1 wt% feed Pep-Cy3), and FRET NP (containing 1 wt% feed Pep-Cy3 and 1.5–9 wt% PLGA-Cy5).

**Figure 4 pharmaceuticals-16-00818-f004:**
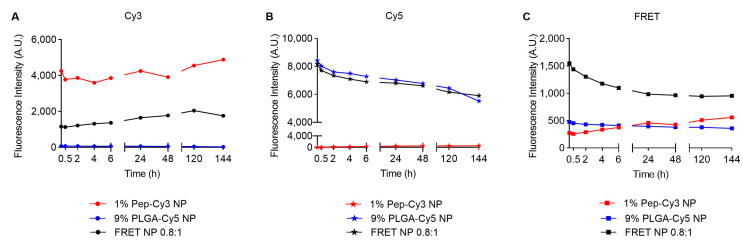
Fluorescence intensity of 0.2 mg/mL 1 % Pep-Cy3 NP, 9% PLGA-Cy5 NP and FRET NP 0.8:1 (1 wt% Pep-Cy3:9 wt% PLGA-Cy5) incubated in PBS (pH 7.4) for 144 h at 37 °C. From left to right: fluorescence intensity of (**A**) Cy3 (λ_ex/em_ = 555/570 nm), (**B**) Cy5 (λ_ex/em_ = 646/662 nm), and (**C**) FRET (λ_ex/em_ = 555/662 nm).

**Figure 5 pharmaceuticals-16-00818-f005:**
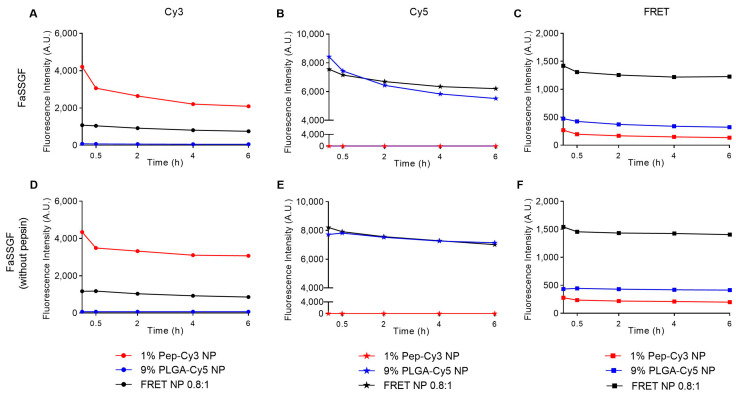
Fluorescence intensity of 0.2 mg/mL 1% Pep-Cy3 NP, 9% PLGA-Cy5 NP, and FRET NP 0.8:1 (1 wt% Pep-Cy3:9 wt% PLGA-Cy5), incubated for 6 h at 37 °C in FaSSGF (pH 1.6, with pepsin) (**A**–**C**) and in FaSSGF (pH 1.6, without pepsin) (**D**–**F**). From left to right: fluorescence intensity of Cy3 (λ_ex/em_ = 555/570 nm), Cy5 (λ_ex/em_ = 646/662 nm), and FRET (λ_ex/em_ = 555/662 nm).

**Figure 6 pharmaceuticals-16-00818-f006:**
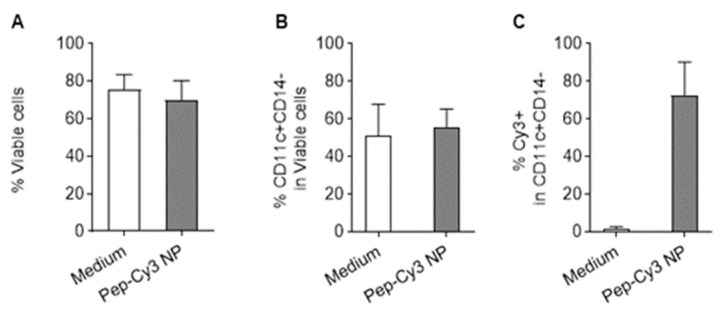
Flow cytometry data of human immature moDC after 2.5 h incubation with 0.06 mg/mL 2% Pep-Cy3 NP at 37 °C. Percentages of viable cells (**A**), CD11c+CD14- human moDC viable cells (**B**), and cellular uptake (Cy3-positive CD11c+CD14- cells) (**C**) were gated using medium-treated human immature moDC as control in the flow cytometry analysis. Data are presented as mean ± SEM, *n* = 3. Human immature moDC were derived and cultured from three different donors.

**Figure 7 pharmaceuticals-16-00818-f007:**
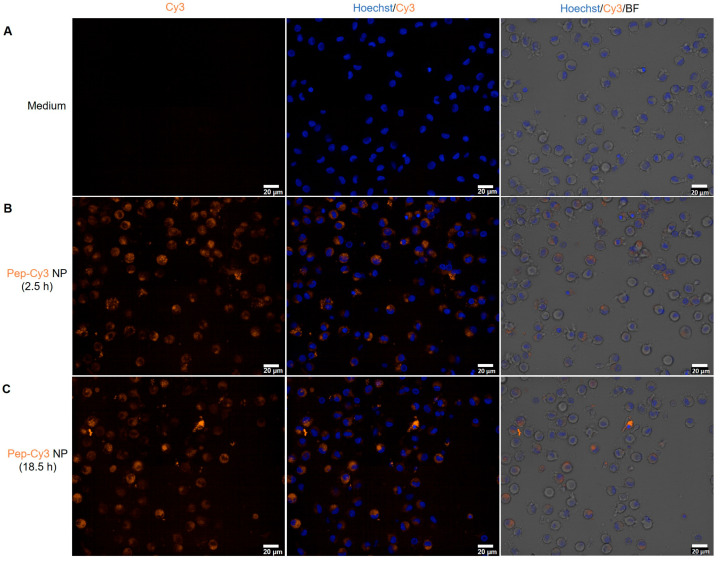
Representative fluorescent microscopy images of cellular uptake of 2% Pep-Cy3 NP (0.06 mg/mL) by day 7 human moDC: medium-treated control (**A**) and internalized 2% Pep-Cy3 NP after (**B**) 2.5 and (**C**) 18.5 h of incubation at 37 °C; bars indicate 20 μm.

**Figure 8 pharmaceuticals-16-00818-f008:**
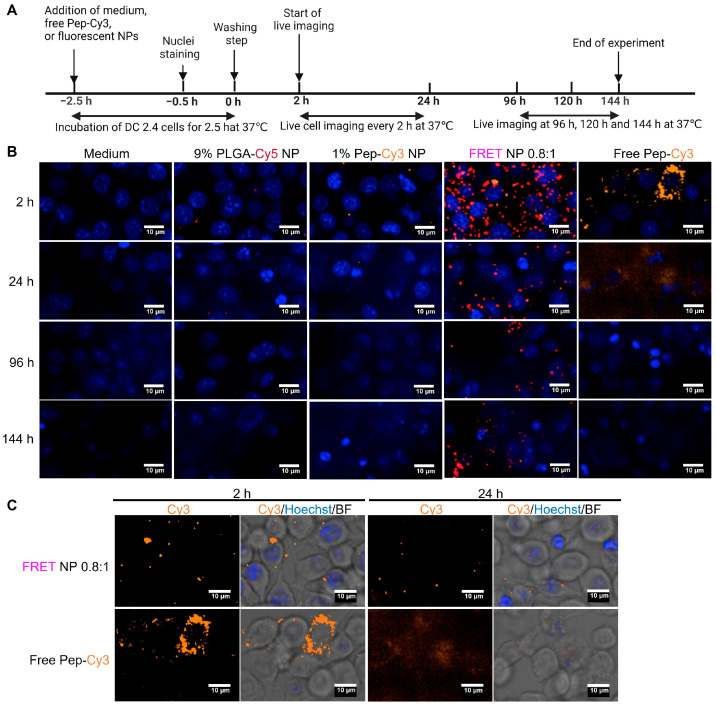
(**A**) Experimental scheme and (**B**) cellular images of internalized fluorescent nanoparticles and free Pep-Cy3 by murine DC 2.4. Cells were washed and live-imaged over 144 h after 2.5 h of incubation of DC 2.4 cells with medium, 9% PLGA-Cy5 NP, 1% Pep-Cy3 NP, FRET NP 0.8:1 (1% Pep-Cy3:9% PLGA-Cy5), or 1 µg/mL free Pep-Cy3. Shown are representative merged confocal fluorescence microscopy images (**B**) of nuclei (Hoechst, blue, λ_ex_ = 405 ± 5 nm), Cy3 (orange, λ_ex_ = 561 ± 2 nm), Cy5 (red, λ_ex_ = 640 + 4/−5 nm), and FRET (λ_ex_ = 561 ± 2 nm) taken 2, 24, 96, and 144 h after the washing step; bars indicate 10 μm. (**C**) Fluorescence from the FRET NP 0.8:1 and free Pep-Cy3 treated DC 2.4 cells in confocal image of Cy3 alone and merged confocal images of Cy3 (orange), nuclei (Hoechst, blue), and bright field (BF, confocal path) 2 (**left** panel) and 24 h (**right** panel) after the washing step, respectively.

**Figure 9 pharmaceuticals-16-00818-f009:**
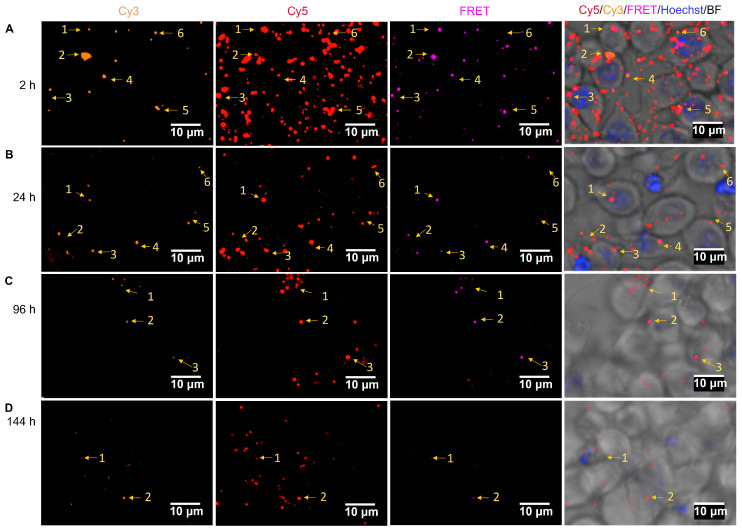
Intracellular localization of FRET NP after internalization by DC 2.4 cells. According to the scheme of Figure 8A, DC 2.4 cells were incubated with FRET NP 0.8:1 (1% Pep-Cy3:9% PLGA-Cy5) for 2.5 h and subsequently washed. Shown are representative confocal fluorescence microscopy images of DC 2.4 cultured with FRET NP (**A**) 2, (**B**) 24, (**C**) 96, and (**D**) 144 h after the washing step; bars indicate 10 μm. Colocalizations of Cy3 (orange, λ_ex_ = 561 ± 2 nm), Cy5 (red, λ_ex_ = 640 + 4/−5 nm), and FRET (purple, λ_ex_ = 561 ± 2 nm) fluorescence are indicated with the yellow arrows and numbers.

**Table 1 pharmaceuticals-16-00818-t001:** Characteristics of Pep-Cy3, PLGA-Cy5 and FRET nanoparticles.

Formulation (*n* = 1)	Nanosight Analysis ^1^	Zetasizer Nano S ^2^		Zeta-Potential (mV)	EE ^3^ (%)	LC ^4^ (%)	Actual Molar Ratio Pep-Cy3: PLGA-Cy5	Yield (%)	E_PR_ ^5^ (%)
	D_h_ (nm)	D_h_ (nm)	PDI						
0.1% Pep-Cy3 NP *	-	616 ± 24	0.15 ± 0.06	−2.5 ± 0.1	-	-		63	-
0.5% Pep-Cy3 NP *	-	463 ± 19	0.22 ± 0.01	−2.1 ± 0.2	31	0.16		82	-
1% Pep-Cy3 NP *	-	338 ± 10	0.15 ± 0.01	−1.9 ± 0.4	31	0.31		73	-
1.5% Pep-Cy3 NP *	-	526 ± 10	0.20 ± 0.04	−1.8 ± 0.1	44	0.66		62	-
2% Pep-Cy3 NP *	-	574 ± 22	0.15 ± 0.02	−2.1 ± 0.2	51	1.02		77	-
Empty PLGA NP	291 ± 5	358 ± 5	0.13 ± 0.01	−2.5 ± 0.4	-	-		73	-
1.5% PLGA-Cy5 NP	330 ± 7	345 ± 5	0.09 ± 0.05	−2.6 ± 0.3	-	-		73	-
2% PLGA-Cy5 NP	304 ± 14	372 ± 9	0.12 ± 0.02	−2.5 ± 0.3	-	-		85	-
4% PLGA-Cy5 NP	309 ± 3	249 ± 6	0.08 ± 0.01	−3.2 ± 0.5	-	-		66	-
7% PLGA-Cy5 NP	361 ± 10	600 ± 6	0.29 ± 0.01	−2.5 ± 1.6	-	-		66	-
9% PLGA-Cy5 NP	170 ± 21	631 ± 5	0.21 ± 0.01	−2.1 ± 0.3	-	-		68	-
1% Pep-Cy3 NP	260 ± 12	253 ± 2	0.08 ± 0.02	−2.7 ± 0.3	30	0.30		56	-
FRET NP 0.8:1 (1% Pep-Cy3: 9% PLGA-Cy5)	227 ± 26	236 ± 3	0.05 ± 0.01	−2.1 ± 0.3	70	0.70	0.56:1	38	60
FRET NP 2:1 (1% Pep-Cy3: 4% PLGA-Cy5)	276 ± 5	240 ± 4	0.10 ± 0.03	−2.5 ± 0.5	24	0.24	0.43:1	48	26
FRET NP 4:1 (1% Pep-Cy3: 2% PLGA-Cy5)	295 ± 8	259 ± 2	0.09 ± 0.02	−2.6 ± 0.4	30	0.30	1.11:1	60	16
FRET NP 5:1 (1% Pep-Cy3: 1.5% PLGA-Cy5)	265 ± 6	217 ± 6	0.07 ± 0.04	−3.7 ± 1.0	16	0.16	0.77:1	58	14

* Formulations marked with * were prepared with 50 mg/mL PLGA in DCM, and the other formulations without * were prepared with 40 mg/mL PLGA in DCM. Percentages mentioned are the weight% of labeled compound within the nanoparticles. ^1^ Measurement of hydrodynamic diameter (D_h_) in quintuplicate. ^2^ Measurement in triplicate (D_h_: hydrodynamic diameter; PDI: Polydispersity Index). ^3^ EE: Encapsulation Efficiency. ^4^ LC: Loading capacity. ^5^ E_PR_: FRET Proximity Ratio (Ratiometric FRET efficiency).

## Data Availability

Data is contained within the article or Appendix A.

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
