# Peer review of "Live Cell Imaging by Förster Resonance Energy Transfer Fluorescence to Study Trafficking of PLGA Nanoparticles and the Release of a Loaded Peptide in Dendritic Cells"

_pharmaceuticals, 2023, doi:10.3390/ph16060818_

Round 1

Reviewer 1 Report

The comments are listed below.

1. Error bars are required in Figures 4 and 5.

2. Quality of the figures requires improvement.

3. Conclusion needs major revision in terms of drawbacks as well as future aspects of present studies.

Overall the research idea is excellent and well executed. needs minor revision.

need minor revision,

Reviewer 2 Report

please find in the attachment

Reviewer 3 Report

I went through the article with the title ``Intracellular trafficking of PLGA nanoparticles and the release of a loaded peptide via live cell imaging of dendritic cells by Förster Resonance Energy Transfer fluorescence". In terms of the findings, the paper presents intriguing possibilities. Overall, based on the research and the data, it is appropriate for publication. However, there are some issues that has to be fixed before publication.

Please make the title more catchy.

Need to revise the keywords.

Please add numerical results in abstract and revise it.

The language of the article should be improved.

There are some typo, space and spelling errors that should be rectified.

Please concern some recent papers for introduction to support the study well.

Need to write the novelty of this work in introduction section.

Please improve the quality of figures.

Please rewrite the conclusion concise and numerical value based.

The language of the article should be improved.

Reviewer 4 Report

The present article entitled “Intracellular trafficking of PLGA nanoparticles and the release of a loaded peptide via live cell imaging of dendritic cells by Förster Resonance Energy Transfer fluorescence" describes the β-lactoglobulin derived tolerogenic peptide (BLG-Pep) loaded in poly(lactic-co-glycolic acid) (PLGA) nanoparticles protected mice against cow milk allergy development. Thus, this reviewer recommends the publication of this work in this Journal after addressing the following concerns.

Comments

1.      In abstract section should be provided quantitative information.

2.      Hypothesis of this work should provide elaborately in the introduction section.

3.      The author should provide reference in the following section. 2.1 Preparation and characterization of the Cy3 labeled b-lactoglobulin derived peptide and Cy5 labeled PLGA, 2.2 Characteristics of single and dual labeled PLGA nanoparticles, and 3.2 Solid-phase peptide synthesis.

4.      The author needs to be performed the study of FTIR, XRD, and TEM analysis of PLGA nanoparticles.

5.      Statistical measurement procedure should be provide at the end of the experimental section.

6.      The conclusion should be revised with outstanding point of this work.

7.      Typographical errors and superfluous spaces throughout the manuscript should be corrected.
